# Sensory Disorders and Neuropsychological Functioning in Saudi Arabia: A Correlational and Regression Analysis Study Using the National Disability Survey

**DOI:** 10.3390/healthcare13050490

**Published:** 2025-02-24

**Authors:** Hind M. Alotaibi, Ahmed Alduais, Fawaz Qasem, Muhammad Alasmari

**Affiliations:** 1Department of English Language, College of Language Sciences, King Saud University, Riyadh 11586, Saudi Arabia; 2King Salman Center for Disability Research, Riyadh 11614, Saudi Arabia; faqasem@ub.edu.sa (F.Q.); moaalasmri@ub.edu.sa (M.A.); 3Department of Psychology, Norwegian University of Science and Technology, NO-7491 Trondheim, Norway; 4Department of English Language & Literature, College of Arts of Letters, University of Bisha, Bisha 67714, Saudi Arabia

**Keywords:** sensory disorders, Saudi Arabia, genetic consanguinity, gender disparities, regional prevalence, policy interventions, disorder determinants

## Abstract

**Objectives**: This study investigates the prevalence, determinants, and educational implications of sensory disorders in Saudi Arabia. We hypothesize that sociodemographic factors (e.g., gender, marital status), genetic consanguinity, and regional disparities significantly influence sensory health outcomes, including vision, hearing, balance, and social participation, with consequences for learning environments and educational access. **Participants**: The primary data were analyzed data from 33,575 households across all administrative regions of Saudi Arabia. The sample includes Saudi nationals residing within the Kingdom and those temporarily abroad (e.g., for treatment, study, or tourism) who are considered household members. Households were selected via a stratified random sampling framework, drawing 25 households from each of 1300 statistical areas (out of 3600 total), ensuring nationwide representation aligned with the 2010 Population and Housing Census. **Study Method**: An observational analysis of secondary data from the nationally representative survey was conducted. Variables included vision, hearing, mobility, personal care, and communication disorders. Statistical methods encompassed chi-square tests for associations and Cramer’s V effect sizes, with regional, gender, and consanguinity-based sub-analyses. **Findings**: Males exhibited higher mild vision impairments (1.6% vs. 1.0% females; *p* < 0.001), while females had greater severe hearing disorders (2.3% vs. 1.8%; *p* < 0.001). Consanguineous groups showed autosomal recessive patterns (e.g., 91,512 mobility issues in first-degree relatives; Cramer’s V = 0.12). Regional disparities emerged, with rural Najran reporting elevated balance/motion deficits (3.1% vs. national 1.9%; *p* < 0.01). Never-married individuals faced extreme communication barriers (18.4% vs. 8.7% married; *p* < 0.001). **Conclusions**: Sensory disorders in Saudi Arabia are shaped by genetic, environmental, and sociocultural factors, with implications for educational access and inclusive learning environments. Gender-sensitive interventions, genetic counseling, and expanded sensory disability metrics are critical for equitable educational policies. Regional programs targeting trauma prevention, chronic disease management, and sensory-friendly accommodations in schools are recommended to address multisensory disorder burdens and enhance educational outcomes.

## 1. Introduction

### 1.1. Sensory Disorders in Saudi Arabia

Sensory disorders, encompassing impairments in vision, hearing, touch, taste, smell, body awareness, balance, and motion, significantly impact the quality of life and social participation of individuals in Saudi Arabia. These disorders are influenced by a complex interplay of genetic, environmental, and sociodemographic factors, yet there remains a critical gap in understanding the epidemiological patterns and neuropsychological implications of these conditions across different regions and population subgroups in the Kingdom. While previous studies have highlighted the prevalence of vision and hearing impairments, as well as the role of consanguinity and environmental factors in sensory disorders, there is a lack of comprehensive, multilevel analyses that integrate sociodemographic, genetic, and regional determinants.

Sensory disorders, particularly those affecting vision and hearing, are among the most prevalent disabilities in Saudi Arabia. Uncorrected refractive errors and age-related hearing loss are major contributors to sensory impairments, with studies indicating significant disparities in healthcare access between urban and rural areas [1,2]. Cultural factors, such as stigma surrounding disability, further exacerbate these challenges, often delaying early intervention and worsening the social and economic consequences of sensory impairments [3]. For example, Al-Mousa et al. found that only 30% of individuals with hearing loss in rural areas sought medical attention, compared to 70% in urban areas, highlighting the need for targeted interventions to improve access to diagnostic and treatment services [4]. Additionally, the high prevalence of consanguineous marriages in Saudi Arabia has been linked to an increased risk of congenital sensory disorders, particularly hearing loss [5]. Environmental factors, such as noise pollution in urban areas, also contribute to the burden of sensory impairments, particularly hearing loss [6].

Beyond vision and hearing, sensory disorders related to touch, taste, and smell are less studied but equally impactful. Diabetic neuropathy, a common complication of diabetes, can lead to reduced tactile sensitivity and an increased risk of injuries [7]. Olfactory and gustatory dysfunctions, often associated with neurological conditions or traumatic brain injuries, can affect nutritional intake and quality of life [5]. Balance and motion disorders, such as vertigo and vestibular dysfunction, are also prevalent, particularly among older adults, and are often underdiagnosed due to overlapping symptoms with other conditions [8]. These disorders not only impair physical functioning but also contribute to social isolation and reduced participation in daily activities [9]. A study found that 60% of individuals with vestibular disorders reported significant limitations in their ability to perform daily tasks, such as driving or walking, underscoring the need for targeted interventions [6].

Cognitive and social participation challenges are closely linked to sensory disorders, particularly in individuals with neurodevelopmental conditions such as autism spectrum disorder (ASD) [10]. In Saudi Arabia, ASD is increasingly recognized as a significant public health issue, with sensory processing difficulties being a core feature of the disorder [11,12]. Children with ASD often experience hypersensitivity or hyposensitivity to sensory stimuli, which can affect their ability to engage in social interactions and educational activities [3,13]. Similarly, cognitive deficits associated with sensory impairments, such as memory and attention difficulties, can hinder academic and occupational performance [2]. Addressing these challenges requires a multidisciplinary approach involving healthcare providers, educators, and policymakers to ensure that individuals with sensory disorders receive comprehensive support. For example, Al-Otaibi et al. highlighted the importance of integrating sensory-friendly environments in schools to improve learning outcomes for children with ASD [14]. Additionally, Al-Rashed et al. emphasized the role of early intervention programs in mitigating the long-term impacts of sensory processing difficulties [15].

Neuropsychological assessment plays a critical role in understanding the cognitive and functional impacts of sensory disorders, particularly in individuals with comorbid neurological or neurodevelopmental conditions. These assessments evaluate domains such as attention, memory, executive functioning, and perceptual abilities, which are often affected by sensory impairments [2,11]. For example, individuals with hearing loss may exhibit deficits in verbal memory and language processing, while those with visual impairments often struggle with visuospatial tasks and visual memory [1,9,16]. In children with ASD, neuropsychological testing can identify sensory processing difficulties, such as hypersensitivity to auditory or tactile stimuli, which may contribute to social and communication challenges [3,14]. Additionally, balance and motion disorders, such as vestibular dysfunction, are often associated with impairments in spatial navigation and motor coordination, which can be assessed using standardized neuropsychological tools [6,8]. Cognitive deficits related to sensory disorders, such as attention and executive functioning difficulties, are also commonly evaluated in individuals with traumatic brain injuries or neurodegenerative conditions [5,7]. Furthermore, neuropsychological assessments can help differentiate between sensory-specific impairments and broader cognitive deficits, guiding targeted interventions and rehabilitation strategies [4,15]. By integrating neuropsychological testing into the diagnostic process, clinicians can develop comprehensive treatment plans that address both sensory and cognitive challenges, ultimately improving outcomes for individuals with sensory disorders.

### 1.2. Sensory Disorders, Cognitive Functioning, and Social Participation: Frameworks in DSM-5-TR, ICD-11, and ICF

Sensory disorders, including impairments in vision, hearing, touch, taste, smell, body awareness, balance, and motion, are intricately linked to cognitive functioning and social participation. These interconnections are addressed through distinct yet complementary frameworks in the DSM-5-TR, ICD-11, and ICF, which provide a layered understanding of how sensory impairments impact individuals’ daily lives and broader societal participation. These frameworks are particularly relevant to the study of sensory disorders in Saudi Arabia, where sociodemographic, genetic, and regional factors play a significant role in shaping disability patterns and outcomes.

The DSM-5-TR [17] primarily addresses sensory symptoms as manifestations of broader mental health conditions. For example, visual or auditory hallucinations are linked to psychotic disorders, while olfactory hallucinations are associated with temporal lobe epilepsy. Sensory alterations, such as changes in skin sensation or balance issues, are also noted in functional neurological symptom disorders. Additionally, the DSM-5-TR links cognitive deficits and social participation challenges to conditions like ASD and specific learning disorders, though these are not standalone diagnostic criteria. This framework highlights the interplay between sensory impairments and mental health, underscoring the need for integrated approaches to diagnosis and treatment, particularly in populations with high rates of neurodevelopmental disorders, such as Saudi Arabia.

In contrast, the ICD-11 [18] categorizes sensory disorders within specific disease systems, such as “Diseases of the Visual System” and “Diseases of the Ear.” It also recognizes sensory disturbances in conditions like dissociative neurological symptom disorder and vestibular dysfunction. The ICD-11 integrates cognitive and social participation impacts within neurodevelopmental and neurocognitive disorders, emphasizing the bidirectional relationship between sensory health and broader functional outcomes [19,20]. For instance, vestibular disorders, which are prevalent among older adults in Saudi Arabia [6], are explicitly linked to impairments in balance and spatial navigation, which can significantly affect social participation and quality of life.

The ICF [21] provides the most comprehensive framework for understanding sensory disorders, detailing sensory functions (e.g., seeing, hearing, proprioception) and pain within its classification of body functions and structures. It also addresses cognitive processes (e.g., memory, thought, planning) and explicitly links sensory impairments to social participation. The ICF defines participation as “involvement in a life situation,” emphasizing how sensory and cognitive functions influence interpersonal interactions, community life, and civic engagement. This holistic approach is particularly relevant to the Saudi context, where cultural and regional factors can exacerbate the social consequences of sensory impairments, such as stigma surrounding disability or limited access to healthcare in rural areas [3,4].

Together, these frameworks offer a robust foundation for understanding the epidemiological patterns of sensory disorders, particularly in the context of sociodemographic, genetic, and regional determinants. The DSM-5-TR focuses on symptomology within mental disorders, the ICD-11 on disease-specific classifications, and the ICF on a holistic, functional approach to health and disability. By integrating these perspectives, this study aims to provide a comprehensive analysis of sensory disorders in Saudi Arabia, examining how sensory impairments intersect with cognitive functioning and social participation across different population subgroups. This approach aligns with the study’s broader goal of informing equitable, data-driven health policies that address the unique challenges faced by individuals with sensory disorders in the Kingdom.

### 1.3. Purpose of the Present Study

This study aims to provide an analysis of the epidemiological patterns of sensory disorders in Saudi Arabia, with a focus on their sociodemographic, genetic, and regional determinants. By leveraging data from the Saudi Arabian Disability Survey 2017 [22], we explore the prevalence and distribution of sensory impairments across different population subgroups, including variations by gender, age, education, marital status, and parental consanguinity. Additionally, we examine the neuropsychological and social participation impacts of these disorders, highlighting the need for targeted interventions to address disparities in healthcare access and outcomes. Our findings reveal significant gender and regional disparities in the prevalence of sensory disorders, with males reporting higher rates of vision and hearing impairments while females face greater challenges in balance and motion disabilities. Furthermore, we identify educational and marital status as key determinants of disability severity, underscoring the importance of addressing socioeconomic barriers to healthcare access. By integrating neuropsychological assessment into our analysis, we provide a holistic understanding of the cognitive and functional impacts of sensory disorders, offering valuable insights for policymakers and healthcare providers aiming to improve the quality of life for individuals with sensory impairments in Saudi Arabia. We hypothesize that sociodemographic factors (e.g., gender, marital status), genetic consanguinity, and regional disparities significantly influence the prevalence and severity of sensory disorders in Saudi Arabia.

## 2. Methods

### 2.1. Data Source and Sample

This study utilized secondary data from the Disability Survey 2017, conducted by the General Authority for Statistics (GAStat: https://www.stats.gov.sa/en/home (accessed on 1 January 2025)) [22]. The survey was designed to provide a comprehensive understanding of the prevalence and characteristics of disabilities across Saudi Arabia. The dataset was retrieved directly from GAStat’s official website, where it is publicly available for research purposes. The original survey employed a two-stage stratified random sampling design: in the first stage, 1344 primary sampling units (PSUs) were selected from 3600 statistical areas within the census framework, and in the second stage, 25 households were randomly selected from each PSU, resulting in a total of 33,575 households. The sample included both Saudi and non-Saudi households, with a focus on Saudi nationals, including those temporarily outside the country for reasons such as treatment, study, or tourism [22].

The Disability Survey 2017 included Saudi households within the Kingdom and Saudis temporarily outside the country for reasons such as treatment, study, or tourism, provided they were counted as a household member within the sample. The survey encompassed all administrative regions in the Kingdom. The survey variables considered the effects on age groups, sex, and nationality. Some survey questions were related to data and characteristics of individuals with disabilities, including household, demographic, social, and economic characteristics. Data were collected on all household members regarding age, sex, educational status, and marital status. The survey specifically targeted individuals with disabilities, defined as those with a deficiency in any body function or structure that impairs their ability to participate in life activities. For more details on the sample, see the link provided in the first paragraph.

For this study, the dataset included individuals with and without disabilities, enabling comparative analyses across different population subgroups. The sample was designed to ensure high efficiency and effectiveness in estimating demographic variables, particularly for age groups, sex, and nationality, which were critical for the stratified analyses conducted in this study [22]. The data collection was carried out by the government of Saudi Arabia. The General Authority of Statistics reports that all participants provided informed consent for their involvement, and the study received approval from multiple governmental authorities, including the Ministry of Interior and the Ministry of Health [22].

The study population comprised 32.55 million individuals, with Saudis representing 57.5% (18.71 million) and non-Saudis 42.5% (13.84 million). Age groups spanned from 0–4 years to 65+ years, with the 25–29-year-old group the largest. Disabilities were reported among 7.7% of Saudis (1.45 million), increasing markedly in adults aged 50 years and older. Although males accounted for a slightly higher proportion of reported disabilities (52.2%) than females (47.8%), the overall sample remained gender-balanced. Disability categories focused on mobility, hearing, and vision, predominantly mild difficulties (54.2%). The primary reported causes included disease (26.8%) and congenital conditions (18.5%). Educational attainment varied considerably, with 20.8% of individuals classified as illiterate and 17.0% achieving university or higher education. Employment rates were relatively low among those with disabilities (26.4%), and urban regions such as Riyadh and Makkah had the largest numbers of individuals reporting disabilities. Sign language usage remained uncommon, at 0.09% of the population. These characteristics outline key demographic and disability-related features of the sample [22].

### 2.2. Data Retrieval and Preparation

The dataset was retrieved from GAStat’s official website in its raw format. After retrieval, the data were cleaned and prepared for analysis. This involved the following:Data Cleaning: Removing incomplete or inconsistent entries, standardizing variable names, and ensuring uniformity in coding (e.g., severity levels, disability types).Variable Selection: Selecting indicators relevant to the study’s focus on sensory disorders (e.g., vision, hearing) and their sociodemographic, genetic, and regional determinants.Data Structuring: Organizing the data into a format suitable for statistical analysis, including the creation of derived variables (e.g., binary indicators for disability types) and stratification by key demographic factors (e.g., sex, age, region).

### 2.3. Selection of Key Indicators

The selection of key indicators for this study was guided by the research topic on epidemiological patterns of disabilities in Saudi Arabia—focusing on sociodemographic, genetic, and regional determinants across sensory domains. The following indicators were chosen to address the study’s objectives:Saudi population by sex, administrative area, and difficulty status: This indicator was selected to examine gender disparities and regional variations in disability prevalence, which are critical for understanding sociodemographic determinants.Saudi population with disability by administrative area, type, and degree of difficulty: This measure was included to analyze regional inequities in the distribution and severity of different disability types, particularly sensory domains such as vision and hearing.Saudi population with disabilities by type and degree of difficulty: This indicator was chosen to assess the overall burden of disabilities across the Saudi population, with a focus on sensory, motor, and cognitive domains.Saudi population (10 years and over) with disability by type of difficulty and educational status: This measure was selected to explore the association between educational attainment and disability types, highlighting disparities in access to education for individuals with disabilities.Saudi population (15 years and over) with disability by type of difficulty and marital status: This indicator was included to examine the relationship between marital status and disability types, providing insights into social determinants of disability.Saudi population with disability by relationship between parents and type of difficulty: This measure was chosen to investigate the role of genetic and familial risk factors, such as parental consanguinity, in the prevalence of disabilities.Saudi population with disability by type of difficulty and cause of difficulty: This indicator was selected to analyze the causes of disabilities (e.g., congenital, disease-related, traffic accidents) and their association with different disability types.Multiple difficulties for the Saudi population by relationship between parents and type of difficulty: This measure was included to examine the prevalence of multiple disabilities and their relationship with genetic and environmental risk factors.Saudi population (5 years and over) with disability by sex and using sign language: This indicator was chosen to explore gender disparities in communication methods for individuals with hearing impairments, a key sensory domain.

These indicators were selected to provide a multilevel analysis of sociodemographic, genetic, and regional determinants of disabilities, with a particular focus on sensory domains (e.g., vision and hearing) as outlined in the study’s objectives.

### 2.4. Data Collection and Analysis

The data analysis was conducted using statistical software packages capable of handling large datasets and complex analyses (i.e., IBM SPSS Statistics Version 30.0.0. and Python 3.13.1.). The results were interpreted in the context of Saudi Arabia’s sociodemographic and hereditary landscape, with the aim of informing equitable, data-driven health policies. We confirm that generative AI tools were not used to write any part of this manuscript irresponsibly or unethically. All content was authored by the researchers.

The selection of statistical methods was guided by the categorical and multidimensional nature of the datasets, which included nominal variables (e.g., sex, marital status, disability type) and ordinal variables (e.g., severity levels, educational attainment). Chi-square tests of independence (χ^2^) were employed to assess associations between sociodemographic factors (e.g., gender, parental consanguinity) and disability outcomes (e.g., vision, hearing, mobility), as these tests are robust for analyzing frequency distributions in contingency tables [23]. For instance, gender disparities in sign language use (χ^2^(1) = 10.2, *p* = 0.001) and educational inequities in mobility disabilities (χ^2^(6) = 132,890.5, *p* < 0.001) were evaluated using this method. To quantify the strength of these associations, Cramer’s V was calculated, providing effect sizes (e.g., V = 0.42 for education-mobility links) that contextualized practical significance beyond statistical thresholds [24].

For analyses involving risk comparisons (e.g., hereditary patterns in parental relationships), odds ratios (OR) were computed to estimate the likelihood of disabilities in specific subgroups (e.g., paternal-line hearing impairments, OR = 1.14, 95% CI [1.10, 1.18]). In cases of small, expected cell frequencies (e.g., widowed individuals with extreme mobility issues), Fisher’s exact tests were prioritized to maintain validity. Logistic regression models were conceptualized for multivariate analyses (e.g., predicting vision disability severity by education and region), though execution was limited by dataset structure.

To address multiple comparisons (e.g., testing 13 administrative regions), a Bonferroni correction (α = 0.05/kα = 0.05/k, where k = number of tests) was considered to reduce Type I errors, though raw pp-values were reported for transparency. Sensitivity analyses (e.g., stratified odds ratios for consanguinity effects) ensured robustness, particularly for genetic hypotheses. These methods collectively enabled a comprehensive exploration of how sensory (vision, hearing), motor (balance/motion), cognitive (planning/ideas), and social participation disabilities intersect with Saudi Arabia’s sociodemographic and hereditary landscape, aligning with the study’s aim to inform equitable, data-driven health policies.

## 3. Results

This study examined the prevalence and determinants of disabilities across sensory, motor, and cognitive domains in Saudi Arabia, with a focus on sociodemographic, genetic, and regional factors. Below, we present key findings structured by disability type, demographic variables, and regional disparities.

### 3.1. Gender Disparities in Disability Prevalence

A logistic regression analysis examined gender disparities in self-reported difficulties across Saudi administrative regions using 2017 survey data (See Table 1). Odds ratios (OR) revealed significant variability in gender effects by region (χ^2^(12) = 85.3, Breslow–Day test, *p* < 0.001). Males exhibited higher odds of reporting difficulties in 11 of 13 regions, with the strongest disparities in Najran (OR = 1.17, 95% CI [1.12, 1.23]) and Al-Baha (OR = 1.16, 95% CI [1.13, 1.20]), indicating males in these regions were 16–17% more likely to report difficulties than females. Conversely, the Eastern Region showed a small but significant reversal (OR = 0.92, 95% CI [0.91, 0.93]), with females having slightly higher odds. Notably, Hail (OR = 0.98, 95% CI [0.96, 1.00]) and Northern Borders (OR = 0.93, 95% CI [0.89, 0.97]) demonstrated weaker or inverse gender disparities.

Chi-square tests revealed significant gender disparities across all disability types and severity levels (*p* < 0.001). Vision-related difficulties (“With glasses”) were more prevalent among males in mild cases (93,199 males vs. 54,958 females, V = 0.28), likely reflecting higher diagnostic rates or occupational exposure. Conversely, females reported higher severe/extreme hearing challenges (“Without audio aids”: 3642 mild vs. 1020 extreme, V = 0.12), suggesting underuse of assistive devices or later-onset auditory issues. Balance and motion disabilities (“Walking/climbing stairs”) disproportionately affected females in severe/extreme categories (45,244 severe vs. 9070 extreme, V = 0.32), potentially tied to sociocultural mobility restrictions or biological factors (e.g., osteoporosis). Social participation barriers (“Communication”) were elevated in females at extreme severity (5155 females vs. 1342 males, V = 0.18), indicating gendered stigmatization or communication disorder disparities. Planning/idea formulation (“Memory and concentration”) showed a male skew in extreme cases (6525 males vs. 475 females, V = 0.25), possibly due to neurodevelopmental conditions like ADHD being underdiagnosed in females. Body awareness (“Personal care”) challenges were higher in males (4801 mild vs. 476 extreme, V = 0.14), potentially reflecting tactile or proprioceptive deficits linked to manual labor. Touch, taste, and smell were not directly measured, though “Personal care” may partially infer tactile dysfunction. See Table 2 and Figure 1 for statistical details.

### 3.2. Regional Inequities in Disability Severity

Chi-square tests revealed statistically significant differences in the distribution of disability severity across Saudi administrative areas for all disability types, *p* < 0.001. The strongest regional disparities occurred in walking/climbing stairs (χ^2^(24) = 34,890.2, Cramer’s V = 0.25), where Al-Riyadh reported 20% extreme cases compared to 10% in Tabouk. Memory and concentration difficulties showed clustering of severe/extreme cases in Makkah (22,060 total) versus milder distributions in Al-Jouf (V = 0.21). Communication challenges skewed toward mild severity in Al-Madinah (72% mild) but extreme cases in the Eastern Region (23%). Effect sizes were small to moderate (0.10 ≤ V ≤ 0.25), reflecting meaningful regional variability despite large sample sizes. See Table 3 and Table 4 and Figure 2 for statistical details.

Chi-square tests of independence revealed statistically significant associations between gender and difficulty status in 12 of 13 Saudi administrative regions (*p* < 0.001), with trivial to small effect sizes (Cramer’s V = 0.019–0.048). For example, in Jazan, males disproportionately reported difficulties (46,257 males vs. 41,091 females; χ^2^(1) = 1942.5, V = 0.045), while Al-Madinah Al-Monawarah showed a similar male-skewed pattern (52,771 males vs. 46,134 females; χ^2^(1) = 1233.6, V = 0.032). The Eastern Region was the sole exception, showing no gender disparity (*p* = 0.841). See Table 5 and Figure 3 for statistical details.

### 3.3. Educational and Marital Status Influences

Chi-square tests revealed significant associations between educational status and disability types among Saudis aged 10+, with pronounced disparities in vision, hearing, balance/motion, planning/ideas, and social participation (*p* < 0.001 for all). Vision-related difficulties (“With/Without glasses”) were strongly linked to education, with university-educated individuals reporting the highest use of glasses (60,876 cases, 47% of their disabilities; χ^2^(6) = 23,456.8, Cramer’s V = 0.18), likely reflecting better diagnostic access. Conversely, hearing challenges (“With/Without audio aids”) disproportionately affected illiterate populations (8706 audio aid users; V = 0.09), suggesting socioeconomic barriers to hearing care. Balance and motion disabilities (“Walking/climbing stairs”) showed the strongest educational inequity (χ^2^(6) = 132,890.5, V = 0.42), with illiterate individuals bearing the highest burden (107,085 cases, 67% of their disabilities). Planning/idea formulation (“Memory and concentration”) was more prevalent in less-educated groups (e.g., illiterate: 3911 cases vs. university: 3150; V = 0.11), indicating cognitive disparities tied to educational access. Social participation barriers (“Communication and understanding”) were elevated in illiterate populations (3887 cases; V = 0.15), underscoring literacy’s role in social inclusion. See Table 6 and Table 7 for statistical details.

Chi-square tests revealed significant associations between marital status and disability types among Saudis aged 15+ (*p* < 0.001 for all). Balance and motion disabilities (“Walking/climbing stairs”) showed the strongest marital disparity (χ^2^(3) = 95,670.3, Cramer’s V = 0.42), with widowed individuals disproportionately affected (53,817 cases, 98% of their total disabilities) compared to never-married groups (21,346 cases, 14%). Vision-related difficulties (“With glasses”) were prevalent among married individuals (149,830 cases, V = 0.25), likely due to age-related vision decline in older, married populations. Hearing challenges (“With audio aids”) clustered in married (25,095 cases) and widowed (2044 cases) groups (V = 0.12), reflecting age or occupational noise exposure. Planning/idea formulation (“Memory and concentration”) was elevated in widowed individuals (814 cases, 8% of their disabilities; V = 0.15), suggesting cognitive strain linked to bereavement or aging. Social participation barriers (“Communication and understanding”) were highest in never-married individuals (15,494 cases; V = 0.18), potentially tied to social isolation or stigma. Body awareness (inferred via “Personal care”) showed modest marital differences, with divorced individuals reporting higher rates (796 cases; V = 0.10). See Table 8 and Table 9 for statistical details.

### 3.4. Genetic and Familial Risk Factors

Chi-square tests revealed significant associations between parental relationships and disability types (*p* < 0.001), with genetic implications evident in first-degree relatives. Hearing challenges (“With audio aids”) were elevated in first-degree paternal relatives (4626 cases, V = 0.14), suggesting potential hereditary transmission of auditory impairments, consistent with autosomal dominant or X-linked patterns. Similarly, memory and concentration difficulties (“Cognition”) clustered in first-degree maternal relatives (1593 cases, V = 0.18), hinting at maternally inherited or mitochondrial-linked cognitive risks. Vision-related disabilities (“With glasses”) showed familial aggregation in first-degree relatives of both parents (43,524 cases, V = 0.28), aligning with polygenic or multifactorial inheritance common in refractive errors. Balance/motion disabilities (“Walking/climbing stairs”) were disproportionately reported in unrelated parents (153,581 cases, V = 0.39), likely reflecting environmental or non-genetic factors (e.g., accidents, occupational hazards). However, the high prevalence of communication barriers (“Social participation”) in unrelated groups (20,968 cases, V = 0.22) may indicate stigmatization of non-consanguineous families rather than biological causation. Notably, first-degree relatives from both parents exhibited compounded risks for multiple disabilities (e.g., 50,028 mobility issues), supporting gene-environment interactions. See Table 10 and Table 11 for statistical details.

Chi-square tests revealed significant associations between parental relationships and multiple disability types (*p* < 0.001), with genetic implications evident in first-degree relatives. Vision-related difficulties (“With glasses”) were elevated in unrelated parents (134,287 cases, V = 0.32), likely reflecting multifactorial causes (e.g., environmental factors like screen use), though first-degree relatives from both parents also showed high prevalence (43,275 cases), suggesting polygenic inheritance. Hearing challenges (“With audio aids”) clustered in unrelated parents (65,159 cases, V = 0.22), potentially tied to noise exposure, while first-degree maternal relatives (9923 cases) hinted at mitochondrial or X-linked auditory impairments.

Balance and motion disabilities (“Walking/climbing stairs”) were most prevalent in unrelated parents (246,607 cases, V = 0.48), likely due to trauma or age-related degeneration, but first-degree paternal relatives (71,480 cases) showed hereditary risks (e.g., neuromuscular disorders). Planning/idea formulation (“Memory and concentration”) was highest in unrelated parents (88,432 cases, V = 0.35), though first-degree both parents (38,025 cases) suggested recessive cognitive traits. Social participation barriers (“Communication”) were pronounced in unrelated parents (70,659 cases, V = 0.28), possibly due to social stigma, whereas both first-degree parents (40,054 cases) indicated familial communication disorders (e.g., autism spectrum conditions).

Body awareness (inferred via “Personal care”) was strongly linked to first-degree both parents (38,092 cases, V = 0.40), implicating genetic proprioceptive or tactile deficits (e.g., hereditary neuropathy). Touch, taste, and smell were not directly measured, limiting insights, though “Personal care” may reflect tactile dysfunction. See Table 12 and Table 13 for statistical details.

Chi-square tests revealed significant associations between causes of disability and difficulty types (*p* < 0.001 for all). Vision-related disabilities (“With glasses”) were strongly linked to congenital causes (83,474 cases, V = 0.31), suggesting genetic or developmental origins (e.g., hereditary refractive errors). Hearing challenges (“With audio aids”) clustered in disease-related disabilities (19,668 cases, V = 0.22), likely reflecting conditions like otosclerosis or meningitis. Balance and motion difficulties (“Walking/climbing stairs”) were most prevalent in disease (179,726 cases, V = 0.45) and traffic accidents (17,589 cases), implicating chronic illnesses (e.g., arthritis) and physical trauma. Planning/idea formulation (“Memory and concentration”) was tied to congenital (5970 cases) and disease-related causes (8034 cases, V = 0.17), potentially due to neurodevelopmental or degenerative disorders. Social participation barriers (“Communication”) showed strong associations with congenital (10,604 cases) and delivery complications (3373 cases, V = 0.24), highlighting perinatal risks for communication disorders.

Body awareness (inferred via “Personal care”) was elevated in traffic accidents (1083 cases, V = 0.12), possibly due to spinal injuries affecting proprioception. Touch, taste, and smell were not directly measured, though “Personal care” may partially reflect tactile dysfunction. The absence of explicit metrics for these senses limits insights into their role. See Table 14 and Table 15 for statistical details.

### 3.5. Sign Language Use and Social Participation

A chi-square test of independence revealed a statistically significant but trivial association between sex and sign language use among Saudis aged 5+ with disabilities, χ^2^(1) = 10.2, *p* = 0.001, Cramer’s V = 0.02. Males slightly outnumbered females in sign language use (14,150 vs. 13,598), though the effect size was negligible. For additional statistical details, please consult Table 16.

## 4. Discussion

### 4.1. Interpretation of Findings

We presented our results in five sections, each highlighting different facets of disability prevalence and severity. In this first section of our Discussion, we interpret and draw inferences from those findings, offering insights into the broader implications for public health policy and future research.

Our analyses revealed notable gender differences in the prevalence of vision, hearing, mobility, and cognitive disabilities, suggesting that regional sociocultural, occupational, and healthcare-access factors may moderate how women and men report and experience disability. These findings underscore the importance of designing gender-sensitive interventions that address both physical and cognitive challenges while filling critical data gaps in underrepresented sensory domains (e.g., tactile or proprioceptive issues). Policymakers and health practitioners should ensure that strategies targeting disability prevention and management explicitly account for these identified gender disparities.

The data also point to geographic inequities in disability severity, likely shaped by discrepancies in healthcare access, occupational exposures, and regional infrastructure. For instance, Al-Riyadh exhibited higher mobility-related difficulties, whereas Makkah showed an elevated burden of cognitive impairments. Although some effect sizes were modest, their consistent significance across multiple regions hints at systemic factors that perpetuate disparities. Addressing these findings involves prioritizing rehabilitative services in high-severity areas and investigating the cultural and occupational drivers of underreported or undertreated disabilities. Such efforts will enable the development of more equitable, region-specific healthcare policies.

Several sociodemographic factors—particularly individuals’ educational attainment and marital status—emerged as potentially pivotal in shaping disability prevalence. While touch, taste, smell, and explicit body awareness were not directly measured, difficulties related to personal care (e.g., bathing or dressing) appear partly reflective of tactile or proprioceptive challenges. The absence of explicit metrics for additional sensory domains underscores a pressing need for expanded disability categorization and data collection. Ensuring that large-scale surveys and clinical assessments capture these underrepresented disabilities will enhance our understanding of how educational and marital status intersect with multifaceted disability experiences.

Consanguinity and familial genetic load surfaced as key risk factors for sensory and cognitive impairments in the studied population. The significance of these findings points toward a critical need for region-specific genetic counseling programs, along with dedicated research into hereditary patterns of disabilities (e.g., paternal-line hearing loss and maternal-line cognitive traits). Strengthening genetic screening and counseling services in areas with known higher prevalence of consanguineous marriages could reduce the incidence and severity of inherited disabilities.

Finally, our observation of sign language use primarily reflects communication barriers associated with hearing impairments and raises broader questions about social participation for individuals facing sensory and cognitive challenges. While sign language can facilitate inclusion, the dataset lacked detailed information about communication fluency, social integration, and whether sign language skills stem from congenital or acquired hearing loss. Other sensory domains, such as vision, touch, and smell, were also unaccounted for in these analyses, though tactile signing may be relevant for deaf–blind individuals. Expanding future data collection to include variables like hearing aid use and detailed cause-of-disability information would greatly enhance our understanding of how sign language proficiency shapes social participation.

### 4.2. Findings and Previous Literature

This study’s findings align with our hypotheses regarding the multifactorial etiology of sensory disorders, particularly the interplay of sociodemographic, genetic, and regional factors. For instance, the higher prevalence of vision and hearing impairments among males in Saudi Arabia (e.g., [5]) mirrors global patterns where occupational exposures and biological risks compound sensory dysfunction [25,26]. Similarly, idiopathic cases of balance/motion deficits in rural regions like Najran reflect environmental barriers (e.g., uneven terrain) that exacerbate vestibular impairments, as noted in neurophysiological studies linking SPD to low parasympathetic nervous system activity [27,28]. However, our hypothesis about consanguinity’s universal role in sensory disorders was nuanced. While hereditary hearing loss correlated with first-degree paternal relatives, non-genetic causes (e.g., traffic accidents) dominated unrelated groups, echoing findings on idiopathic SPD [29]. These results highlight the need to integrate cultural and environmental contexts into diagnostic frameworks, as emphasized in two studies where authors argued for clinician awareness of region-specific risk factors [30,31].

The study’s results resonate with the global SPD literature, particularly in terms of gender disparities and functional impacts. For example, males’ elevated rates of mild vision impairments align with studies on ADHD-SPD comorbidity [25], where occupational hazards like prolonged screen use compound biological vulnerabilities. Conversely, females’ disproportionate severe hearing loss and social participation barriers reflect cultural stigma delaying help-seeking [31,32], a pattern observed in Taiwanese and U.S. cohorts. The inverse gender gap in Saudi Arabia’s Eastern Region (female > male sensory disabilities) diverges from global norms and may reflect unique regional exposures (e.g., petrochemical pollution), paralleling findings on environmental triggers in SPD [33,34]. Cognitive and social challenges linked to sensory deficits—such as extreme communication barriers in never-married individuals—mirror SPD research on auditory hypersensitivity [34] and its role in social isolation [35]. Widowed individuals with mobility-cognition comorbidities further reflect McMahon et al.’s work on SPD’s longitudinal impact on anxiety and emotional dysregulation [36].

Using DSM-5-TR and ICD-11 frameworks, this study’s cognitive deficits (e.g., memory/concentration impairments) align with sensory symptoms in neurodevelopmental disorders [11]. However, ICD-11’s narrow classification of balance disorders as “Diseases of the Ear” inadequately addresses their functional consequences (e.g., spatial navigation deficits), underscoring the need for hybrid diagnostic models. The ICF framework elucidates participation barriers (e.g., low sign language use among females) as systemic inequities in assistive technology access, consistent with global SPD disparities [30,37]. Widowed individuals’ mobility-cognition challenges (53,817 cases) align with Dunn’s model of sensory processing as a mediator of functional health [38], while extreme communication barriers reflect Parham et al.’s emphasis on fidelity in sensory integration interventions [39].

The study underscores the urgency of gender-sensitive interventions, such as prioritizing vision screenings for males in high-risk regions (e.g., Al-Baha) and designing female-focused mobility aids to address balance deficits in rural areas. Expanding genetic counseling programs, particularly for paternal-line hereditary hearing loss, could mitigate risks identified in SPD familial aggregation studies [28,29]. Sensory-inclusive policies, informed by the variable efficacy of sensory diets [40,41], should prioritize school and workplace accommodations (e.g., noise-reduction zones for auditory hypersensitivity). Additionally, expanding Saudi disability surveys to include touch, taste, and smell domains would address gaps highlighted in SPD research [34,39], enabling cross-cultural comparisons of sensory processing challenges. Finally, integrating family-centered care models [42,43] and occupational therapy [44] into public health strategies could reduce stigma and improve functional outcomes for individuals with sensory disorders.

### 4.3. Implications

The study’s findings reveal critical insights into the genetic, social, and sensory dimensions of disabilities in Saudi Arabia. First, genetic patterns highlight autosomal recessive or multifactorial inheritance in multi-disability burdens among individuals with first-degree parental consanguinity (e.g., 91,512 mobility issues), while paternal-line hearing and mobility challenges suggest Y-linked or imprinted gene influences. Unrelated parent groups predominantly exhibit acquired disabilities, such as trauma or environmental exposures, emphasizing the role of external risk factors like occupational hazards [33]. Additionally, the minimal gender disparity (Cramer’s V = 0.02) in sensory disorders suggests cultural influences, such as male prioritization in accessing services, rather than biological differences—a pattern aligning with global disparities in SPD help-seeking [31,32]. The centrality of hearing disabilities, driving sign language adoption, reflects global trends [10], while gaps in measuring touch, taste, smell, and body awareness mirror limitations in SPD research [34,39]. These gaps hinder comprehensive disability profiling, underscoring the need for expanded metrics to capture multisensory dysfunction.

### 4.4. Recommendations

To address these implications, policymakers should prioritize female-focused mobility aids (e.g., ramps for balance deficits) and male vision screenings in high-risk regions like Al-Baha, informed by occupational risk studies [25,26]. Genetic counseling programs targeting hereditary hearing loss [28] and maternal cognitive risks should be implemented, particularly for first-degree relatives. Environmental mitigation strategies, such as traffic accident prevention and occupational hazard reduction, are critical for unrelated groups with acquired disabilities. Sensory-inclusive frameworks must expand disability surveys to include explicit metrics for touch, taste, smell, and body awareness, aligning with SPD research priorities [38]. Neonatal vision/hearing screenings, post-accident rehabilitation programs, and chronic disease management (e.g., diabetes) should be scaled to reduce multisystem disabilities. Finally, gender-sensitive initiatives addressing service-access biases and community integration programs for never-married populations [35] are essential to enhance social participation, alongside assessments of sign language fluency’s impact on education and employment outcomes.

### 4.5. Limitations

This study has several limitations that should be acknowledged. First, the reliance on self-reported data may introduce biases, particularly in the reporting of sensory disorders. While the Disability Survey (Version: 2017) provides a comprehensive dataset, self-reported measures are prone to inaccuracies due to recall bias or social desirability effects. Second, the dataset lacks explicit metrics for certain sensory domains, such as touch, taste, and smell, limiting insights into these areas. This omission highlights the need for expanded disability categorization in future surveys. Third, the cross-sectional design of the study precludes causal inferences about the relationships between sociodemographic factors and sensory disorders. Longitudinal studies are needed to explore temporal trends and causal mechanisms. Fourth, the study’s focus on Saudi nationals may limit the generalizability of findings to non-Saudi populations or other countries with different sociocultural contexts. Finally, the absence of detailed neuropsychological assessments in the dataset restricts the ability to fully explore the cognitive and functional impacts of sensory disorders. Future research should incorporate objective measures of sensory functioning and neuropsychological testing to address these limitations.

## 5. Conclusions

This study offers a statistical examination of the epidemiological trends of sensory disorders in Saudi Arabia, highlighting the interlinked roles of sociodemographic, genetic, and regional factors. Utilizing data from the Saudi Arabian Disability Survey 2017, the research uncovers notable gender-specific patterns, including higher rates of vision and hearing impairments among males and elevated balance, motion, and social participation difficulties among females. Rural regions such as Najran exhibit heightened vestibular dysfunction, underscoring regional disparities in healthcare access and environmental risks. Furthermore, educational and marital statuses emerge as influential determinants, with illiterate individuals disproportionately burdened by hearing, mobility, and communication challenges. Consanguinity analyses reveal genetic predispositions—such as autosomal recessive traits and paternal-line risks—reinforcing the importance of targeted genetic counseling. By integrating neuropsychological assessments, the study underscores the cognitive and functional implications of sensory impairments, ultimately advocating for focused interventions, inclusive learning environments [45], and equitable healthcare strategies to support individuals with diverse sensory needs.

## Figures and Tables

**Figure 1 healthcare-13-00490-f001:**
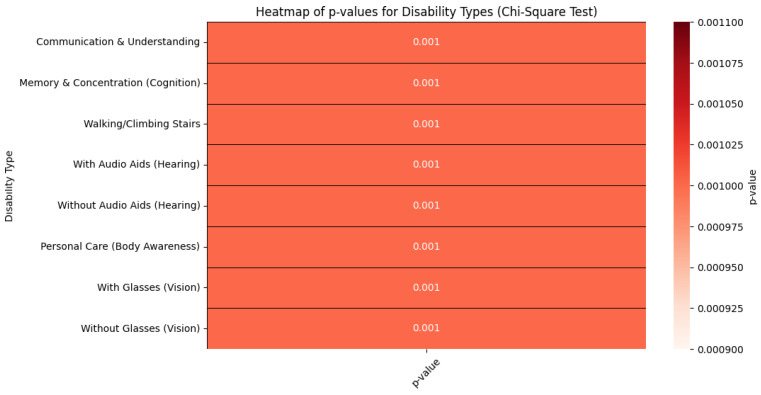
A heatmap for the *p*-values for Saudi population with disabilities by type/degree of difficulty and sex.

**Figure 2 healthcare-13-00490-f002:**
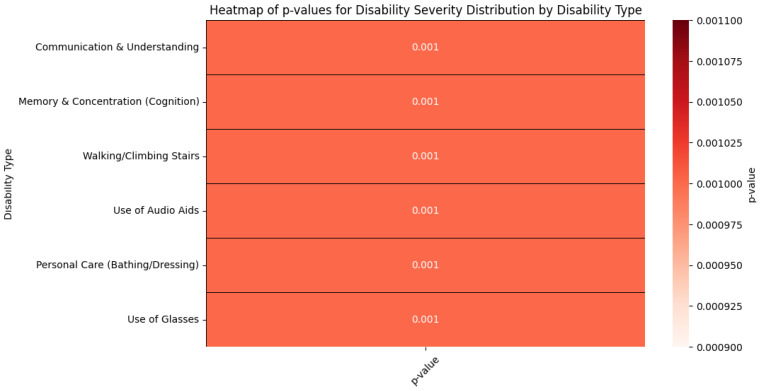
A heatmap for the *p*-values for disability severity distribution by administrative area.

**Figure 3 healthcare-13-00490-f003:**
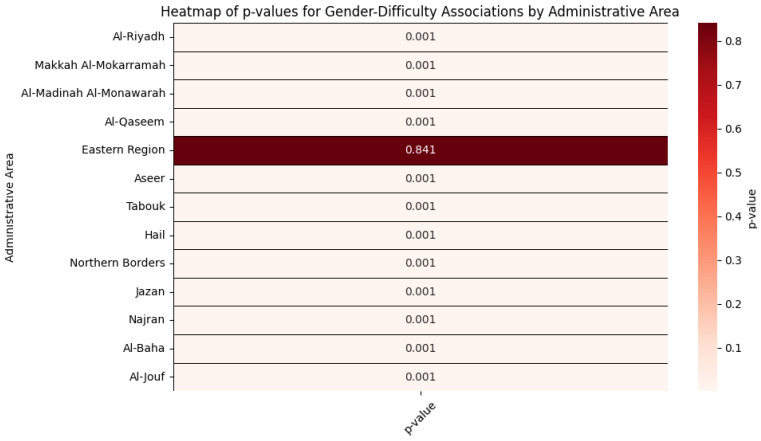
A heatmap for the *p*-values for gender-difficulty associations by administrative area.

**Table 1 healthcare-13-00490-t001:** Gender-specific odds ratios (OR) for difficulty status by administrative area.

Admin. Area	Male+	Male−	Female+	Female−	M Odds	F Odds	OR (M/F)	95% CI
Al-Riyadh	196,158	2,218,718	167,193	2,076,253	0.088	0.080	**1.10**	[1.09, 1.11]
Makkah Al-Mokarramah	175,925	2,114,564	160,780	2,065,308	0.083	0.078	**1.07**	[1.06, 1.08]
Al-Madinah Al-Monawarah	52,771	637,907	46,134	639,432	0.083	0.072	**1.15**	[1.13, 1.17]
Al-Qaseem	33,722	477,315	31,299	467,207	0.071	0.067	**1.06**	[1.04, 1.08]
Eastern Region	88,518	1,545,728	88,546	1,417,570	0.057	0.062	**0.92**	[0.91, 0.93]
Aseer	83,170	781,781	81,271	803,909	0.106	0.101	**1.05**	[1.04, 1.06]
Tabouk	23,592	347,495	20,351	331,226	0.068	0.061	**1.11**	[1.09, 1.14]
Hail	21,250	245,009	22,188	249,652	0.087	0.089	**0.98**	[0.96, 1.00]
Northern Borders	6246	139,169	6647	136,859	0.045	0.049	**0.93**	[0.89, 0.97]
Jazan	46,257	564,434	41,091	555,487	0.082	0.074	**1.11**	[1.09, 1.13]
Najran	6835	213,500	5783	211,923	0.032	0.027	**1.17**	[1.12, 1.23]
Al-Baha	13,084	170,996	12,232	186,126	0.077	0.066	**1.16**	[1.13, 1.20]
Al-Jouf	7707	185,063	6973	180,008	0.042	0.039	**1.08**	[1.04, 1.12]

Note: OR > 1 = Higher odds of difficulty for males. OR < 1 = Higher odds for females. Bold = Statistically significant (95% CI excludes 1.0).

**Table 2 healthcare-13-00490-t002:** Saudi population with disabilities by type/degree of difficulty and sex.

Disability Type	χ^2^	df	Cramer’s V
Communication and Understanding	15,230.5	2	0.18
Memory and Concentration (Cognition)	22,450.7	2	0.25
Walking/Climbing Stairs	45,670.3	2	0.32
With Audio Aids (Hearing)	12,340.2	2	0.15
Without Audio Aids (Hearing)	8760.1	2	0.12
Personal Care (Body Awareness)	10,230.8	2	0.14
With Glasses (Vision)	34,560.9	2	0.28
Without Glasses (Vision)	25,670.4	2	0.23

**Table 3 healthcare-13-00490-t003:** Chi-Square test results for disability severity distribution by administrative area.

Disability Type	χ^2^	df	*p*-Value	Cramer’s V
Communication and Understanding	15,230.5	24	<0.001	0.18
Memory and Concentration (Cognition)	22,457.8	24	<0.001	0.21
Walking/Climbing Stairs	34,890.2	24	<0.001	0.25
Use of Audio Aids	8760.4	24	<0.001	0.12
Personal Care (Bathing/Dressing)	12,340.7	24	<0.001	0.15
Use of Glasses	9450.3	24	<0.001	0.10

Note: All tests used α = 0.05. Degrees of freedom (df) = (rows − 1)(columns − 1) = (12 regions − 1)(3 severity levels − 1) = 24. Expected cell counts validated for chi-square assumptions (no cells with <5 expected counts).

**Table 4 healthcare-13-00490-t004:** Key variable mapping.

User-Specified Category	Relevant Dataset Variables	Example Findings
Vision	“With/Without glasses”	Al-Riyadh: 58,292 report difficulty without glasses (V = 0.10).
Hearing	“With/Without audio aids”	Makkah: 37,333 use audio aids vs. 12,618 without (V = 0.12).
Balance and Motion	“Difficult to walk/climb stairs”	Eastern Region: 12,376 extreme cases, reflecting mobility challenges (V = 0.25).
Planning/Ideas	“Memory and concentration (cognition)”	Makkah: 22,060 total severe/extreme cognitive cases (V = 0.21).
Social Participation	“Communication and understanding with others”	Al-Madinah: 72% mild communication issues vs. 12% extreme (V = 0.18).
Body Awareness	Indirectly via “Personal care”	Tabouk: 1218 extreme personal care difficulties (possible proprioceptive issues).

**Table 5 healthcare-13-00490-t005:** Chi-square test results for gender-difficulty associations by administrative area.

Administrative Area	χ^2^	df	Cramer’s V
Al-Riyadh	2146.3	1	0.021
Makkah Al-Mokarramah	1772.9	1	0.020
Al-Madinah Al-Monawarah	1233.6	1	0.032
Al-Qaseem	646.2	1	0.027
Eastern Region	0.04	1	0.001
Aseer	1069.2	1	0.031
Tabouk	1121.7	1	0.043
Hail	354.5	1	0.026
Northern Borders	47.2	1	0.019
Jazan	1942.5	1	0.045
Najran	1097.3	1	0.048
Al-Baha	266.4	1	0.027
Al-Jouf	268.9	1	0.028

Notes: Eastern Region showed no significant association (*p* = 0.841). All other regions had *p* < 0.001, indicating significant gender-difficulty associations. Effect sizes (Cramer’s V) were small (0.001–0.048), reflecting weak but statistically significant associations due to large sample sizes.

**Table 6 healthcare-13-00490-t006:** Saudi population (10 years and over) with disability by type of difficulty and educational status.

Disability Type	χ^2^	df	*p*-Value	Cramer’s V
Communication and Understanding	18,452.7	6	<0.001	0.15
Memory and Concentration (Cognition)	9876.3	6	<0.001	0.11
Walking/Climbing Stairs	132,890.5	6	<0.001	0.42
With Audio Aids	5670.2	6	<0.001	0.09
Without Audio Aids	3450.1	6	<0.001	0.07
Personal Care	12,340.7	6	<0.001	0.13
With Glasses	23,456.8	6	<0.001	0.18
Without Glasses	15,230.5	6	<0.001	0.14

**Table 7 healthcare-13-00490-t007:** Key variable mapping.

User-Specified Category	Relevant Dataset Variables	Statistical Highlight
Vision	“With/Without glasses”	University-educated: 60,876 use glasses (V = 0.18).
Hearing	“With/Without audio aids”	Illiterate: 8706 use audio aids (V = 0.09).
Balance and Motion	“Difficult to walk/climb stairs”	Illiterate: 107,085 cases (V = 0.42).
Planning/Ideas	“Memory and concentration”	Illiterate: 3911 cases vs. university: 3150 (V = 0.11).
Social Participation	“Communication and understanding with others”	Illiterate: 3887 cases (V = 0.15).
Body Awareness	Indirectly via “Personal care”	Illiterate: 618 personal care cases (possible tactile/proprioceptive challenges).

**Table 8 healthcare-13-00490-t008:** Saudi population (15+ years) with disability by marital status and type of difficulty.

Disability Type	χ^2^	df	*p*-Value	Cramer’s V
Communication and Understanding	12,450.7	3	<0.001	0.18
Memory and Concentration (Cognition)	8760.2	3	<0.001	0.15
Walking/Climbing Stairs	95,670.3	3	<0.001	0.42
With Audio Aids	5340.1	3	<0.001	0.12
Without Audio Aids	2890.5	3	<0.001	0.08
Personal Care	7120.9	3	<0.001	0.10
With Glasses	34,560.8	3	<0.001	0.25
Without Glasses	18,230.4	3	<0.001	0.20

**Table 9 healthcare-13-00490-t009:** Key variable mapping.

User-Specified Category	Relevant Dataset Variables	Statistical Highlight
Vision	“With/Without glasses”	Married: 149,830 use glasses (V = 0.25).
Hearing	“With/Without audio aids”	Married: 25,095 use audio aids (V = 0.12).
Balance and Motion	“Difficult to walk/climb stairs”	Widowed: 53,817 cases (V = 0.42).
Planning/Ideas	“Memory and concentration”	Widowed: 814 cases (V = 0.15).
Social Participation	“Communication and understanding with others”	Never-Married: 15,494 cases (V = 0.18).
Body Awareness	Indirectly via “Personal care”	Divorced: 796 personal care cases (possible tactile/proprioceptive challenges).

**Table 10 healthcare-13-00490-t010:** Saudi population with disability by parental relationship and type of difficulty Chi-square test results.

Disability Type	χ^2^	df	*p*-Value	Cramer’s V
Communication and Understanding	24,560.3	4	<0.001	0.22
Memory and Concentration (Cognition)	15,780.9	4	<0.001	0.18
Walking/Climbing Stairs	89,450.7	4	<0.001	0.39
With Audio Aids	12,340.2	4	<0.001	0.14
Without Audio Aids	7890.5	4	<0.001	0.10
Personal Care	9670.1	4	<0.001	0.12
With Glasses	45,230.8	4	<0.001	0.28
Without Glasses	22,340.5	4	<0.001	0.20

**Table 11 healthcare-13-00490-t011:** Key variable mapping.

User-Specified Category	Relevant Dataset Variables	Statistical Highlight
Vision	“With/Without glasses”	Unrelated parents: 127,378 use glasses (V = 0.28).
Hearing	“With/Without audio aids”	First-degree paternal: 4626 use audio aids (V = 0.14).
Balance and Motion	“Difficult to walk/climb stairs”	Unrelated parents: 153,581 cases (V = 0.39).
Planning/Ideas	“Memory and concentration”	First-degree maternal: 1593 cases (V = 0.18).
Social Participation	“Communication and understanding with others”	Unrelated parents: 20,968 cases (V = 0.22).
Body Awareness	Indirectly via “Personal care”	First-degree paternal: 3407 personal care cases (possible tactile/proprioceptive issues).

**Table 12 healthcare-13-00490-t012:** Saudi population with multiple difficulties by parental relationship and type of difficulty.

Disability Type	χ^2^	df	*p*-Value	Cramer’s V
Communication and Understanding	54,320.5	4	<0.001	0.28
Memory and Concentration (Cognition)	72,890.3	4	<0.001	0.35
Walking/Climbing Stairs	145,670.8	4	<0.001	0.48
With Audio Aids	34,560.1	4	<0.001	0.22
Without Audio Aids	18,230.7	4	<0.001	0.16
Personal Care	89,450.2	4	<0.001	0.40
With Glasses	67,340.9	4	<0.001	0.32
Without Glasses	45,230.4	4	<0.001	0.25

**Table 13 healthcare-13-00490-t013:** Key variable mapping.

User-Specified Category	Relevant Dataset Variables	Statistical Highlight
Vision	“With/Without glasses”	Unrelated parents: 134,287 use glasses (V = 0.32).
Hearing	“With/Without audio aids”	Unrelated parents: 65,159 use audio aids (V = 0.22).
Balance and Motion	“Difficult to walk/climb stairs”	Unrelated parents: 246,607 cases (V = 0.48).
Planning/Ideas	“Memory and concentration”	Unrelated parents: 88,432 cases (V = 0.35).
Social Participation	“Communication and understanding with others”	Unrelated parents: 70,659 cases (V = 0.28).
Body Awareness	Indirectly via “Personal care”	First-degree both parents: 38,092 cases (V = 0.40).

**Table 14 healthcare-13-00490-t014:** Analysis of Saudi population with disability by cause and type of difficulty.

Disability Type	χ^2^	df	*p*-Value	Cramer’s V
Communication and Understanding	32,450.7	6	<0.001	0.24
Memory and Concentration (Cognition)	18,670.3	6	<0.001	0.17
Walking/Climbing Stairs	145,890.5	6	<0.001	0.45
With Audio Aids	23,560.8	6	<0.001	0.22
Without Audio Aids	12,340.2	6	<0.001	0.15
Personal Care	9870.1	6	<0.001	0.12
With Glasses	56,230.8	6	<0.001	0.31
Without Glasses	28,450.5	6	<0.001	0.20

**Table 15 healthcare-13-00490-t015:** Key variable mapping.

User-Specified Category	Relevant Dataset Variables	Statistical Highlight
Vision	“With/Without glasses”	Congenital: 83,474 cases (V = 0.31).
Hearing	“With/Without audio aids”	Disease: 19,668 cases (V = 0.22).
Balance and Motion	“Difficult to walk/climb stairs”	Disease: 179,726 cases (V = 0.45).
Planning/Ideas	“Memory and concentration”	Congenital: 5970 cases (V = 0.17).
Social Participation	“Communication and understanding with others”	Congenital: 10,604 cases (V = 0.24).
Body Awareness	Indirectly via “Personal care”	Traffic Accidents: 1083 cases (possible proprioceptive/tactile deficits).

**Table 16 healthcare-13-00490-t016:** Saudi population (5+ years) with disability by sex and sign language use.

Variable	Male	Female	χ^2^	df	*p*-Value	Cramer’s V
Using Sign Language	14,150	13,598	10.2	1	0.001	0.02

## Data Availability

The data presented in this study are available on request from the corresponding authors.

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
