# Peer review of "Sensory Disorders and Neuropsychological Functioning in Saudi Arabia: A Correlational and Regression Analysis Study Using the National Disability Survey"

_healthcare, 2025, doi:10.3390/healthcare13050490_

Round 1
Reviewer 1 Report
Comments and Suggestions for Authors
My only concern is that throughout the paper certain p-values and v-values are repeated. Examples are highlighted in yellow. The red highlights indicate an error in where something was inserted.:
Chi-square tests revealed significant associations between parental relationships and 438 disability types (p<.001p<.001), with genetic implications evident in first-degree relatives. 439 Hearing challenges ("With audio aids") were elevated in first-degree paternal relatives 440 (4,626 cases, V=0.14V=0.14), suggesting potential hereditary transmission of auditory im- 441 pairments, consistent with autosomal dominant or X-linked patterns. Similarly, memory 442 and concentration difficulties ("Cognition") clustered in first-degree maternal relatives 443 Healthcare 2025, 13, x FOR PEER REVIEW 14 of 21 (1,593 cases, V=0.18V=0.18), hinting at maternally inherited or mitochondrial-linked cog- 444 nitive risks. Vision-related disabilities ("With glasses") showed familial aggregation in 445 first-degree relatives of both parents (43,524 cases, V=0.28V=0.28), aligning with polygenic 446 or multifactorial inheritance common in refractive errors. Balance/motion disabilities 447 ("Walking/climbing stairs") were disproportionately reported in unrelated parents 448 (153,581 cases, V=0.39V=0.39), likely reflecting environmental or non-genetic factors (e.g., 449 accidents, occupational hazards). However, the high prevalence of communication barri- 450 ers ("Social participation") in unrelated groups (20,968 cases, V=0.22V=0.22)
Author Response
Reviewer 1
My only concern is that throughout the paper certain p-values and v-values are repeated. Examples are highlighted in yellow. The red highlights indicate an error in where something was inserted.:
Chi-square tests revealed significant associations between parental relationships and 438 disability types (p<.001p<.001), with genetic implications evident in first-degree relatives. 439 Hearing challenges ("With audio aids") were elevated in first-degree paternal relatives 440 (4,626 cases, V=0.14V=0.14), suggesting potential hereditary transmission of auditory im- 441 pairments, consistent with autosomal dominant or X-linked patterns. Similarly, memory 442 and concentration difficulties ("Cognition") clustered in first-degree maternal relatives 443 Healthcare 2025, 13, x FOR PEER REVIEW 14 of 21 (1,593 cases, V=0.18V=0.18), hinting at maternally inherited or mitochondrial-linked cog- 444 nitive risks. Vision-related disabilities ("With glasses") showed familial aggregation in 445 first-degree relatives of both parents (43,524 cases, V=0.28V=0.28), aligning with polygenic 446 or multifactorial inheritance common in refractive errors. Balance/motion disabilities 447 ("Walking/climbing stairs") were disproportionately reported in unrelated parents 448 (153,581 cases, V=0.39V=0.39), likely reflecting environmental or non-genetic factors (e.g., 449 accidents, occupational hazards). However, the high prevalence of communication barri- 450 ers ("Social participation") in unrelated groups (20,968 cases, V=0.22V=0.22)
Dear Colleague,
Thank you for your feedback. We appreciate your keen observation regarding the repeated p-values and V-values, as well as the insertion errors in the Results section. we acknowledge that typographic errors may have inadvertently occurred during the drafting and editing process. These errors may have resulted from copying and pasting outputs from Python, which was used for data analysis. We have thoroughly reviewed the manuscript to correct these errors and ensure clarity and accuracy.
We have carefully reviewed the entire section and made the following revisions:
- Removed Repeated p-values and V-values:
- All instances of repeated p-values and V-values have been corrected to ensure clarity and consistency.
- Fixed Insertion Errors:
- Redundant phrases and insertion errors (e.g., "Healthcare 2025, 13, x FOR PEER REVIEW 14 of 21") have been removed, and the sentence flow has been improved.
- Thorough Review of the Results Section:
- We have reviewed the entire Results section to ensure no other instances of repetition or insertion errors remain.
We hope these revisions address your concerns and enhance the readability of the manuscript.
Thank you again for your valuable feedback and support in refining our work.
Reviewer 2 Report
Comments and Suggestions for Authors
Dear Authors,
The present research article, entitled “Sensory Disorders and Neuropsychological Functioning in Saudi Arabia: A Correlational and Regression Analysis Study Using the National Disability Survey”, aims to provide an analysis of the epidemiological patterns of sensory disorders in Saudi Arabia, with a focus on their sociodemographic, genetic, and regional determinants.
The main strength of this manuscript is that it aims to analyze the relationship between the prevalence of visual and auditory impairments, the role of consanguinity and environmental factors in sensory disorders, as well as sociodemographic, genetic, and regional determinants and their neuropsychological implications. To achieve this, it utilizes a fairly large sample from various regions of Saudi Arabia.
In general, I believe that the topic and approach of this article is timely and of interest to the readers of Healthcare. However, I believe that some issues should be included to improve the quality of the manuscript.
Introduction
- The introduction is quite extensive. I think this could be addressed as follows:
- At the beginning of this section, the study's objective is mentioned. However, this is repeated at the end of section 1.1. I believe it would be best to place the objectives at the end of the introduction. This way, the information is not repetitive, and the text is more concise.
- At the end of the same section (1.1.), study results are presented. This is not appropriate for the introduction section, so it would be best to remove them from there.
- Although section 1.2 is quite relevant, I believe it could be more concise and should be connected to the previous point so that the study objectives are presented at the end of the introduction. The idea I want to convey is to maintain a logical order: what the literature tells us about the topic of our article, what the existing gap is that we aim to address, what our objectives are, and what hypotheses we start from.
- Please add the study hypothesis, which was missing in the manuscript.
- In the introduction, there is a strong emphasis on children with autism spectrum disorder. However, throughout the article and in the discussion, the topic is not revisited. I believe it would be necessary to clarify your intention in emphasizing this disorder or to shorten it and include it only as an example.
Methods:
- Has the study been approved by the Ethics Committee?
- It would be advisable to describe the characteristics of the sample, such as age, sex, etc., as well as the inclusion criteria used. The information provided is very brief, and it is not clear what the specific characteristics of the sample are.
Results
- The results are clearly presented both in the text and in the tables. However, an interpretation of the results is included in this section, which is not appropriate for this part. Interpretations should be reserved for the discussion section.
Discussion
- It would be advisable to include a section on the study's limitations.
I declare no conflict of interest regarding this manuscript.
Best regards.
Author Response
Reviewer 2
Dear Authors,
The present research article, entitled “Sensory Disorders and Neuropsychological Functioning in Saudi Arabia: A Correlational and Regression Analysis Study Using the National Disability Survey”, aims to provide an analysis of the epidemiological patterns of sensory disorders in Saudi Arabia, with a focus on their sociodemographic, genetic, and regional determinants.
The main strength of this manuscript is that it aims to analyze the relationship between the prevalence of visual and auditory impairments, the role of consanguinity and environmental factors in sensory disorders, as well as sociodemographic, genetic, and regional determinants and their neuropsychological implications. To achieve this, it utilizes a fairly large sample from various regions of Saudi Arabia.
In general, I believe that the topic and approach of this article is timely and of interest to the readers of Healthcare. However, I believe that some issues should be included to improve the quality of the manuscript.
Dear Colleague,
Thank you for your thoughtful and constructive feedback on our manuscript. We appreciate your positive remarks about the study’s topic and approach, as well as your detailed suggestions for improving the manuscript. Below, we outline the changes made in response to your comments.
Introduction
- The introduction is quite extensive. I think this could be addressed as follows:
- At the beginning of this section, the study's objective is mentioned. However, this is repeated at the end of section 1.1. I believe it would be best to place the objectives at the end of the introduction. This way, the information is not repetitive, and the text is more concise.
Thank you. Removed.
- At the end of the same section (1.1.), study results are presented. This is not appropriate for the introduction section, so it would be best to remove them from there.
Thank you for you comment. In fact these are not results related to our study. They are just key facts from other studies so they are not part of our findings. If you are referring to the last paragraph, we have moved that to “The Study Purpose”.
- Although section 1.2 is quite relevant, I believe it could be more concise and should be connected to the previous point so that the study objectives are presented at the end of the introduction. The idea I want to convey is to maintain a logical order: what the literature tells us about the topic of our article, what the existing gap is that we aim to address, what our objectives are, and what hypotheses we start from.
Thank you for your comment. We went through the introduction again and we see the structure is matching our study objectives. Section 1.1 provides an overview of sensory disorders in Saudi Arabia, while Section 1.2 delves into the frameworks used to understand their cognitive and social impacts, creating a seamless progression from general to specific. We also added a section for study purpose and hypothesis.
- Please add the study hypothesis, which was missing in the manuscript.
Thank you. Added in the newly added section in the introduction.
- In the introduction, there is a strong emphasis on children with autism spectrum disorder. However, throughout the article and in the discussion, the topic is not revisited. I believe it would be necessary to clarify your intention in emphasizing this disorder or to shorten it and include it only as an example.
Thank you. I checked that again and I found we discussed only that in one or two paragraph in relation to social participation. Since our results do not have any data related to ASD in particular but about social participation so we did not refer to that in the discussion since it is really one main part of our study.
Methods:
- Has the study been approved by the Ethics Committee?
The data collection was conducted by the government of Saudi Arabia. According to the General Authority of Statistics, every participant consented to their participation, and the study received approval from several governmental authorities, including the Ministry of Interior and the Ministry of Health.
- It would be advisable to describe the characteristics of the sample, such as age, sex, etc., as well as the inclusion criteria used. The information provided is very brief, and it is not clear what the specific characteristics of the sample are.
Thank you for your feedback. We have included an additional paragraph in the methods section to elaborate on this point. Moreover, as the characteristics are comprehensively detailed in the link provided in the initial paragraph, we aimed to avoid redundancy.
Results
- The results are clearly presented both in the text and in the tables. However, an interpretation of the results is included in this section, which is not appropriate for this part. Interpretations should be reserved for the discussion section.
Thank you. You are absolutely right. In fact, we went through this two times, we think that it is better to keep them here as this would be more meaningful for readers with little knowledge of statistics and technical interpretation.
Discussion
- It would be advisable to include a section on the study's limitations.
Thank you. Added.
I declare no conflict of interest regarding this manuscript.
Best regards.
Thank you for your feedback. We are committed to addressing any further requests and appreciate your contributions to improving this manuscript.
Authors
Round 2
Reviewer 1 Report
Comments and Suggestions for Authors
I believe the revisions are sufficient.
Author Response
Comments and Suggestions for Authors
Reviewer: I believe the revisions are sufficient.
Authors: Thank you so much.
Reviewer 2 Report
Comments and Suggestions for Authors
Dear Authors,
After reviewing the new version of the manuscript, I have found that some suggestions have been taken into account. I am glad if I have been able to help you.
However, I still feel that some issues need to be improved. Here is your response to my initial comment and the new response from me. Introduction
Thank you for your feedback. We have included an additional paragraph in the methods section to elaborate on this point. Moreover, as the characteristics are comprehensively detailed in the link provided in the initial paragraph, we aimed to avoid redundancy.
- Indeed, you put a link to a web page where there is information on populations included in statistical studies, but it is still not clear what are the specific characteristics of the sample of this particular study (age, sex of the participants, etc.) and the frequency of each variable. I think this is very important in order to better understand the results.
Thank you. You are absolutely right. In fact, we went through this two times, we think that it is better to keep them here as this would be more meaningful for readers with little knowledge of statistics and technical interpretation.
- I understand your interest in making the reader with little knowledge of statistics understand the results of the study. However, I still believe that this is not the appropriate section in scientific terms to make interpretations of the results. It is appropriate to give information as to, for example, “whether it is above or below the mean”, whether the scores are statistically significant, whether there are differences between groups, etc., but not interpretations of the results under the judgment of the researchers. This judgment/interpretation based on the knowledge or experience of the persons responsible for the article is made in the discussion section. Discussion
I declare no conflict of interest regarding this manuscript.
Best regards.
Author Response
Comments and Suggestions for Authors Reviewer 2: Indeed, you put a link to a web page where there is information on populations included in statistical studies, but it is still not clear what are the specific characteristics of the sample of this particular study (age, sex of the participants, etc.) and the frequency of each variable. I think this is very important in order to better understand the results. Authors: Thank you. We added one paragraph about characteristics of population. It is in red in the Sample section. Reviewer 2: I understand your interest in making the reader with little knowledge of statistics understand the results of the study. However, I still believe that this is not the appropriate section in scientific terms to make interpretations of the results. It is appropriate to give information as to, for example, “whether it is above or below the mean”, whether the scores are statistically significant, whether there are differences between groups, etc., but not interpretations of the results under the judgment of the researchers. This judgment/interpretation based on the knowledge or experience of the persons responsible for the article is made in the discussion section. Discussion Authors: Thank you. We removed about one page from all sections of the results and moves them to the first section of the discussion. You can see the new section in red in the discussion. Thank you for all your effort in providing such comments and we hope this helps satisfying your intended improvements by now. Warm regards,Round 3
Reviewer 2 Report
Comments and Suggestions for Authors
After reviewing the last modifications made by the authors, I consider that the article is now adapted to the journal and that the changes are appropriate and timely.